# A Novel Approach for the Fabrication of 3D-Printed Dental Membrane Scaffolds including Antimicrobial Pomegranate Extract

**DOI:** 10.3390/pharmaceutics15030737

**Published:** 2023-02-22

**Authors:** Hatice Karabulut, Songul Ulag, Basak Dalbayrak, Elif Damla Arisan, Turgut Taskin, Mehmet Mucahit Guncu, Burak Aksu, Alireza Valanezhad, Oguzhan Gunduz

**Affiliations:** 1Institute of Pure and Applied Sciences, Metallurgical and Materials Engineering, Marmara University, Istanbul 34722, Turkey; 2Center for Nanotechnology & Biomaterials Application and Research, Marmara University, Istanbul 34722, Turkey; 3Department of Metallurgy and Materials Engineering, Faculty of Technology, Marmara University, Istanbul 34722, Turkey; 4Institute of Biotechnology, Gebze Technical University, Kocaeli 41400, Turkey; 5Department of Pharmacognosy, Faculty of Pharmacy, Marmara University, Istanbul 34668, Turkey; 6Institute of Health Sciences, Department of Microbiology, Marmara University, Istanbul 34854, Turkey; 7Department of Medical Microbiology, School of Medicine, Marmara University, Istanbul 34854, Turkey; 8Department of Dental and Biomedical Materials Science, Nagasaki University Graduate School of Biomedical Science, Nagasaki 852-8588, Japan

**Keywords:** dental membrane, polyvinylalcohol, pomegranate, starch, scaffolds, 3D printing

## Abstract

In this study, a dental membrane scaffold was fabricated using a 3D printing technique, and the antimicrobial effect of pomegranate seed and peel extract were investigated. For the production of the dental membrane scaffold, a combination of polyvinyl alcohol, starch, and pomegranate seed and peel extracts was used. The aim of the scaffold was to cover the damaged area and aid in the healing process. This can be achieved due to the high antimicrobial and antioxidant content of pomegranate seed and peel extracts (PPE: PSE). Moreover, the addition of starch and PPE: PSE improved the biocompatibility of the scaffold, and their biocompatibility was tested using human gingival fibroblast (HGF) cells. The addition of PPE: PSE into the scaffolds resulted in a significant antimicrobial effect on *S. aureus* and *E. faecalis* bacteria. Moreover, different concentrations of starch (1%, 2%, 3% *w*/*v*) and pomegranate peel and seed extract (3%, 5%, 7%, 9%, and 11% PE *v*/*v*) were analyzed to obtain the ideal dental membrane structure. The optimum starch concentration was chosen as 2% *w*/*v* due to it giving the scaffold the highest mechanical tensile strength (23.8607 ± 4.0796 MPa). The pore sizes of each scaffold were studied by SEM analysis, and pore sizes were arranged between 155.86 and 280.96 μm without any plugging problems. Pomegranate seed and peel extracts were obtained by applying the standard extraction method. High-performance liquid chromatography was performed using the diode-array detection (HPLC-DAD) technique to analyze the phenolic content of the pomegranate seed and peel extracts. Two phenolic components of the pomegranate seed and peel extracts were investigated in the following amounts: fumaric acid (17.56 μg analyte/mg extract) and quinic acid (18.79 μg analyte/mg extract) in pomegranate seed extract and fumaric acid (26.95 μg analyte/mg extract) and quinic acid (33.79 μg analyte/mg extract) in pomegranate peel extract.

## 1. Introduction

Much research is being conducted to improve the quality of human life and extend the human lifespan, particularly in the fields of tissue engineering and dental applications. One of the most common problems in dental health care is periodontal disease [1,2,3]. In particular, chronic inflammatory disorders, also known as periodontitis, must be treated to avoid complex inflammations in the affected area and even tooth loss [4]. Hence, novel regenerative techniques are clearly required to encourage oral and periodontal tissue formation and recovery [5,6], and a barrier membrane is used to treat the affected area and prevent the gingival tissue from spreading to the injured area [7].

In addition, as a result of different pathological issues, the insertion of bone grafts under oral skin, or damaged periodontal skin, the affected area needs to be replaced and repaired with other soft tissue [8,9]. This process aims to reconstruct the functional dental membrane barrier that will later be placed under the oral skin and separate the skin from the bone. Additionally, when selecting the ideal dental membrane, this barrier membrane must have good mechanical strength and feasible tear strength and needs to be stable during oral function. Furthermore, the cells need to be attached and differentiated in the dental barrier membrane. In addition to these primary properties, the dental membrane structure plays a critical role in providing blood supply with the necessary nutrients and oxygen to cells. Thus, the dental membrane’s physical and chemical structures need to be improved to treat the damaged area [10,11,12].

Dental membrane products must be biocompatible and biomedically stable to ensure proper periodontal ligament function. The dental membrane’s pores need to be the optimal size to maintain essential conditions for cell attachment and growth [13,14,15,16]. In the literature, there are different types of commercial dental membrane products and other kinds of materials, especially collagen-based products, that have been developed for dental membrane production. However, collagen cannot provide good mechanical strength or stability when implanted in the affected area [6,17,18]. More sophisticated scaffold systems have recently been created to transmit bioactive indicators for periodontium regeneration. In addition, they can replace new tissues after their degradation [19,20]. Furthermore, resorbable dental membranes are suitable for treating the infected area since they do not require extra surgery for their removal. The vast majority of studies have highlighted the necessity of developing advanced membranes that can deliver vital biomolecules such as antibiotics, growth factors, and stem cells while meeting the minimum requirements of good mechanical qualities and degradability [21,22].

Polyvinyl alcohol (PVA) is a water-soluble polymer that has been used for biomedical applications due to its excellent biodegradability, high biocompatibility, and high processability and also because it improves the mechanical strength of the scaffolds. PVA can provide the desired membrane structure, and PVA is preferred for dental membrane production [23,24]. It has been reported that PVA can increase the scaffold’s tensile strength and prevents adhesion and fast attachment to the epithelium [25] and thus it can be used for the outer layer of dental membranes.

Starch is a natural polymer that can also be used in biomedical applications due to its biodegradability with non-toxic by-products, thus promoting cell growth, biocompatibility, and abundant availability. This natural polymer is the second most abundant material in the world. Still, starch cannot be used in its pure form because of its brittleness, low moisture properties, high hydrophilicity, low process control, and thermal instability. With the addition of other materials to starch, it is easy to overcome the disadvantages of starch’s application [26,27,28].

Recently, there has been a need to use alternative materials to toxic chemical materials in biomedical applications, especially since toxic by-products can occur due to the degradation of polyester [29]. Hence, plant sources are a significant natural alternative due to their abundance and high biological activities. Pomegranate extract has long been used as a functional food and folk medicine due to its high antioxidant and antimicrobial content [27,28]. Phenolic compunds are a group of secondary plant metabolites that are often present in a variety of higher plant organs, including fruits, vegetables, cereals, spices, legumes, and nuts. Phenolic compounds have recently been discovered to have a variety of effects, including antioxidant, antimicrobial, anticarcinogenic, and anti-inflammatory activites, and to prevent cardiovascular diseases, cancers, diabetes, and oxidative stress-related diseases. The bioactive significance of pomegranate extract is becoming more widely recognized as a result of extensive research findings. The biological functions of plants are also known to be mediated by phenolic compounds As a result, the HPLC-DAD system was utilized to assess both the qualitative and quantitative characteristics of the phenolic compounds present in the pomegranate extract used in this investigation [30]. Several studies have also shown that pomegranate peel extracts and seed extracts can be applied in different fields such as pharmaceuticals, tissue engineering, dental applications, and food industries. Moreover, it has been reported that pomegranate peel extract has a higher antimicrobial effect than pomegranate seed extract [31,32,33].

Another significant aspect to consider is the availability of several different methods to fabricate dental membranes. One of the methods used to fabricate dental barrier membranes is using 3D printing technology [8,18]. Three-dimensional printing technology is a preferred method of production because of its different advantages, such as the ease of printing, the fabrication of printed products in a large array of designs and sizes, and the ability to create an unlimited number of identically fabricated structures [34,35,36].

Since unwanted periodontal pathogens can resist the oral uptake of antibiotics and form subgingival biofilms in the injured area, it is needed to treat the injured area with a functional dental barrier membrane [37]. Therefore, in this study, a combination of different materials, namely pomegranate peel extract, pomegranate seed extract, starch, and polyvinyl alcohol, has been used to fabricate scaffolds to be applied to the dental membrane. The 3D-printed scaffolds aimed to cover, heal, and protect damaged areas due to the high antimicrobial content of the pomegranate seed and peel extracts.

## 2. Materials and Method

### 2.1. Materials

Polyvinyl alcohol (MW = 89,000–98,000) was purchased from Sigma Aldrich, (St. Louis, MO, USA). Soluble starch was obtained from Carlo Erba. Glutaraldehyde (GA) solution (50%) in water was purchased from Sigma Aldrich, Co., St. Louis, MO, USA.

### 2.2. Preparation of the Pomegranate Peel and Seed Extracts

Pomegranates (*Punica granatum*) were purchased from a local market in Istanbul, Turkey. The pomegranate exocarp, mesocarp, and seeds were manually separated and washed once with water. Pomegranate seeds were compressed in a laboratory porcelain mortar and passed through filter paper to obtain refined fruit juice. The pomegranate peel was cut into small pieces, approximately 1 cm^2^, with scissors. The net amount of the peel used was 100 g; it was boiled in 100 mL of distilled water for 4 h. The peel extract was also filtered using filter paper. The collected solution from the peel and seeds was mixed at the 1:1 (% *v*/*v*) ratio, stirred on the magnetic stirrer for 15 min, and stored at 4 °C for subsequent analysis.

### 2.3. Preparation of the Solutions

Next, 13% PVA (*w*/*v*) was dissolved in distilled water at 90 °C for 2 h, and different concentrations of starch (1% (*w*/*v*), 2% (*w*/*v*) and 3% (*w*/*v*)) were added to the 13% (*w*/*v*) PVA solutions. Each blend of solutions was mixed for 2 h to obtain homogeneous solutions. Pomegranate peel:seed extract (PPE: PSE) was added to the 2% (*w*/*v*) starch and 13% PVA (*w*/*v*) solutions at different ratios of 3%, 5% 7%, 9%, and %11 (*v*/*v*). Solutions at each concentration were transferred into a 10 mL syringe which was connected to an 3D extrusion printer (Hyrel 3D, SDS-5 Extruder, Norcross, GA, USA).

### 2.4. Design and Fabrication of the 3D-Printed Scaffolds

The scaffold was designed with 20 mm × 20 mm dimensions. The scaffold’s configuration was converted to G-codes using the Simplify 3D program. Process parameters were easily controlled using Solidworks. The flow rate of each extract solution was adjusted to 1 mL/h, the printing rate was 10 mm/s, and the nozzle diameter was 0.160 mm (30 gauge). The printed scaffolds had 8 layers and each scaffold was printed at room temperature (25 °C).

### 2.5. Preparation of the Crosslinked 3D Scaffolds

Each dried 3D-printed scaffold was placed onfilter paper, and approximately 5 mL of 50% of GA solution was added to glass Petri dishes. After inserting the glass Petri dishes into the desiccator, filter paper was placed over the desiccator, which was then put into an incubator at 60 °C for 3 h. After the reaction finished, the scaffolds were taken out and stored for further analysis. The exact process has been applied to all scaffolds, including those including PPE: PSE, and the crosslinking temperature was changed from 60 °C to 40 °C.

### 2.6. Characterisation of the Solutions and 3D-Printed Scaffolds

#### 2.6.1. Physical Characterisations of the Solutions

The density and pH of each solution were analyzed at room temperature in triplicate. The density of the solutions was performed using DIN ISO 3507-Gay-Lussac (Boru Cam Inc., Istanbul, Türkiye) in a standard 10 mL density bottle. After device calibration, the pH value of all the solutions was measured with a pH meter (HANNA HI 2020-02 edge).

#### 2.6.2. Morphological Characterisations of the Scaffolds

Scaffold morphology was characterised using SEM (EVO LS 10, ZEISS, Oberkochen, Germany). Pore size and scaffold morphology were observed, and 20 average pore sizes of each scaffold were statistically analyzed using an image analysis software (Olympus AnalySIS, Rockville, MD, USA).

#### 2.6.3. Chemical Characterisations of the Scaffolds

Chemical characterisation and molecular structure of 3D scaffolds and PPE: PSE were carried out by applying FT-IR (FTIR, 4700 Jasco, Tokyo, Japan) at a scanning range of 4000–400 cm^−1^ and resolution of 4 cm^−1^.

#### 2.6.4. Thermal Characterisations of the Scaffolds

Differential Scanning Calorimetry (DSC) (Shimadzu, Koyoto, Japan) was used to identify the thermal behaviour of the scaffolds. Each scaffold and PPE: PSE was placed in an aluminium pan at a temperature ranging from 25 °C to 400 °C. Thermal glass transition temperature (T_g_) and melting temperature (T_m_) were analyzed at a scanning rate of 10 °C/min.

#### 2.6.5. Pomegranate Extract Content Analysis by HPLC-DAD

For the content analysis of pomegranate extracts, pomegranate seed and peel extracts were separately dried in an incubator at 40 °C for 24 h. The content analysis was performed on the dried extracts using an HPLC-DAD instrument (Agilent 1260 Infinity). Therefore, a C18 reverse-phase Nova-Pak (3.9 mm × 150 mm inner diameter, 5 μm) analytical column was used for separation. The temperature of the column was set to 30 °C. Water (0.05% acetic acid) and (B) acetonitrile were used for the mobile phase of the chromatography. The active content analysis of the extracts was carried out using an HPLC-DAD instrument (Agilent 1260 Infinity).

The gradient elution step was used: the mobile phase B was increased from 0% to 20% in 5 min, 40% in 10, 50% in 20, 60% in, 90% in 40, and 20% in 45 min. Re-equilibration of the column was carried out within 10 min. The injection volume was settled as 20 μL. Authentic standards of two compounds were used to develop the analytical method. Ten-millimeter stock solutions were prepared at a 1000 μg/mL concentration. Standard solutions were prepared using several different solvents, and it was determined that the most suitable solvent was methanol for stock solutions and mobile phase A for standard solutions. Before injecting all standards, a 0.45 µm injector tip was filtered through the filter, and 20 µL of the stock solution was injected into the HPLC system. The LOD and LOQ values of the method reported in this study depended on the calibration curve generated from five measurements. The limit of detection (LOD) and limit of quantitation (LOQ) values were calculated according to the following Equations (1) and (2):(1)LOD=Mean +3×Standard deviation
(2)LOQ=Mean +10×Standard deviation.

#### 2.6.6. Mechanical Properties of the Scaffolds

After the scaffold thickness was measured using a digital micrometer (Mitutoyo MTI Corp., CO, USA), tensile tests were applied using a tensile testing device (Shimadzu EZ-LX, Kyoto, Japan). Each scaffold’s lower and upper portions were placed horizontally between the jaws of the device, and 0.1 N of force was applied. The test speed was kept at 5 mm/min. Each group of scaffold tests was assessed in triplicates to characterize the mechanical properties.

#### 2.6.7. In Vitro Release Studies from the 3D-Printed Scaffolds

After each sample of 3D printed scaffolds was cut into small pieces (10 mm × 10 mm) and immersed in 1 mL of PBS, samples were incubated at 37 °C in a thermal shaker (BIOSAN TS-100) at 360 rpm. The release profile of the PPE: PSE was measured at different time intervals of 15 min, 30 min, and 60 min, followed by 1-h intervals. After these time intervals, the PBS was measured with a spectrophotometer UV-1280 (SHIMADZU, 190–600 nm). After each measurement, 1 mL of fresh PBS was added to the current scaffolds for the next measurement. To calculate the concentrations of the released solution, a graph was used with a standard calibration curve. The determined concentrations of PPE: PSE (0.1, 0.2, 0.4, 0.8, and 1 μg/mL) were prepared and measured with a UV-1280 spectrophotometer (SHIMADZU, 190–600 nm). The absorbance graph was determined using the highest absorbance values from the calibration curve.

#### 2.6.8. Cell Culture Assays

Before cell culture assays, all the scaffolds were sterilised under Class II Laminar flow for 90 min. After applying UV sterilization, the scaffolds were seeded onto a 96-well plate with 20 µL DMEM.

##### Cell Culture

Human Gingival Fibroblast (HGF) cells were incubated in a humidified CO_2_ incubator (Thermo) in 10% Fetal Bovine Serum (Gibco) and 1% penicillin-streptomycin (PAN, Biotech), including DMEM high glucose with 4.5 g/L D-Glucose, L-Glutamine, and sodium pyruvate (NutriCulture, Ecobiotech). Cells were thawed with complete DMEM and seeded on the flask. After cells reached 80% confluency, the cells were passaged using trypsin/EDTA (PAN Biotech). Collected cells were centrifuged at 1500 rpm for 5 min. Before centrifugation, cells were counted with a Haemocytometer (Marienfield). The supernatant was discarded, the cell pellet was dissolved with an appropriate amount of medium, and cells were seeded on a 96-well plate using sterile materials at 2500 cells/well for 150 µL/well. This process was repeated for the 1- and 3-day treatments.

##### Cell Viability-MTT Assay

After the 7-day treatment was completed, 10 µL MTT ((3-(4,5-Dimethylthiazol-2-yl)-2,5-Diphenyltetrazolium Bromide)) reagent was added onto the existing medium and incubated for 4 h at 37 °C with 5% CO_2_ to achieve the reduced formazan crystals in the living cells. When the incubation time was completed, the existing medium was exchanged with DMSO (dimethyl sulfoxide), and the plate was placed on the shaker in a dark environment at room temperature. The absorbance values were taken at 570 nm with VarioSkan Plate Reader (Thermo Fisher Scientific, Waltham, MA, USA). Data were analyzed using two-way ANOVA Multiple Comparison Tukey’s tests. Statistical significance was defined as *p* < 0.05.

##### Fluorescence Imaging

After the 7-day cellular treatment, the existing medium was exchanged with a staining solution containing 4 nM DIOC6 (3,3′-Dihexyloxacarbocyanine Iodide) and 2 mg/mL PI (propidium iodide) to observe mitochondria and dead cells. DIOC6 is a dye which is used to examine the mitochondria in the cells at low concentrations, and the PI is used to stain dead cells. Cells were incubated for 15 min before taking fluorescence images via a ZOE fluorescent cell imager (BIORAD) in the appropriate excitation/emission wavelengths.

#### 2.6.9. Antimicrobial Activities of the 3D Scaffolds

*S. aureus* ATCC^®^ 6538™, *C. albicans* ATCC^®^ 5314™, and *E. faecalis* ATCC^®^ 29212™ cell suspensions were cultured overnight in Mueller–Hinton broth, which was adjusted to a 0.5 McFarland turbidity standard (1–2 × 10^8^ CFU/mL). Cultured cell suspensions were inoculated on Mueller–Hinton broth agar plates using an automated plate inoculator. Each scaffold was cut into small circles (5 mm in diameter), and each side of the scaffolds was sterilised for 30 min under UV light (254 nm). Following that, the sterilised disks were placed on the surface of the cell inoculated Mueller–Hinton broth agar plates with the help of sterile forceps. Each plate was incubated at 37 °C for 18 h, and growth inhibition zones around the disks were measured and recorded. Disks containing 2 µg of ampicillin (AMP) and chlorhexidine gluconate (20%) were used for positive control.

## 3. Results and Discussions

In this study, the 3D printing method was used to fabricate functional scaffolds to be used as dental membranes. Polyvinyl alcohol and starch were used as the main components of the scaffolds. To produce an antimicrobial effect against microorganisms in the oral biota, antimicrobial pomegranate peel and seed extracts (PPE: PSE) were used in a ratio of 1:1. Based on previous studies, it was observed that 13% PVA concentration was optimal for printing scaffolds [38]. The effects of different concentrations of the starch and PPE: PSE mixture on 3D-printed scaffolds were observed and discussed. The physical characteristics of each solution were assessed and the density and pH values of the solutions are shown in Table 1. The pH and density of the pure PVA polymer solution were measured at 6.83 and 1.038 g/mL, respectively. However, the addition of starch to a pure PVA solution slowly decreased the pH and density. In addition, at the highest starch concentration, the pH and density of the solution were 5.49 and 1.021 g/mL, respectively. Furthermore, as PPE: PSE were added into control group (13% PVA/2S), the pH and density of the solution gradually decreased. Due to an increase in the PPE: PSE in the control group, the pH and density of the solutions also rapidly dropped. On the other hand, although there was no significant change in the pH value of pomegranate peel extract (PPE) and pomegranate seed extract (PSE), the pH values were observed at 3.48 for PPE, which was higher than the values of the PSE solution at 3.63. Moreover, the densities of pomegranate seed and peel extracts were quite similar and measured at 1.024 and 1.001 g/mL, respectively. According to all results, it is possible to suggest that low pH values of solutions and PPE: PSE can be preferable for their higher antimicrobial effect and storage of the phenolic component concentration in extracts under better conditions [39].

The morphological analysis of the 3D-printed scaffolds was performed using a scanning electron microscope (SEM). The morphological analysis and pore size distributions of each scaffold can be seen in Figure 1 and Figure 2. As it can be seen in Figure 1 and Figure 2, the morphological structure of the 3D-printed scaffolds was altered with the addition of starch and PPE:PSE, and all scaffolds showed great and visible pore structure without any clogging. Additionally, the scaffolds did not show a very significant difference in the mean value of the pore size and structure. When the mean pore sizes of each scaffold were compared, the mean value of the pore size of the pure PVA scaffold showed the lowest pore size compared to scaffolds with only starch added, which was 177.60 µm. The mean value of the pure PVA scaffold increased to 184.99 µm when starch was added at the lowest concentration. In addition to that, when the starch concentration was increased to 2%, the mean value was increased to 232.42 µm. However, when the starch concentration was highest, the mean value of the pore size of the scaffold decreased to 201.27 µm. Although there was no discernible difference in the pore sizes of the scaffolds, the addition of starch to the pure PVA scaffolds did not result in a significant change in the pore size and morphological structure of the pure PVA scaffolds.

Furthermore, the morphologies of the scaffolds changed with the addition of PPE: PSE to the control group. The addition of PPE: PSE decreased the pore size of the scaffolds, as shown in Figure 2. When the PPE:PSE concentration was the lowest, the mean value of the pore sizes of the control group decreased to 160.64 µm. Only scaffolds made from 9% PE solution did not have smaller pores. The scaffolds with the lowest and highest mean pore sizes, at 155.86 µm and 280.96 µm, were found to be 13% PVA/5% PE and 13% PVA/9% PE, respectively. It is known that the scaffold pore size plays a critical role, resulting in the occurrence of undesired contamination in dental membranes with large pore sizes. In addition, due to the small pore size of the dental membrane, cell attachment and growth are difficult to achieve. Since there is no optimal pore size for dental membrane scaffolds, the scaffold pore size used in our study has been chosen to be similar to the pore size of commercial dental membrane scaffolds, which vary in size between 0.2 and 1700 µm [40,41,42,43]. Additionally, according to the study of Ciara et al., the pore size of the scaffolds plays a crucial role in cell attachment, adhesion, proliferation, migration, as well as the formation of the extracellular matrix, and they observed that the pore size of the scaffold was less than 325 µm, which was within the preferred range for cells [44]. Therefore, the pore sizes of fabricated 3D-printed scaffolds can be highly suitable for cell attachment, migration, adhesion, and proliferation in order to mimic the extracellular matrix.

For the chemical analysis of each scaffold, FT-IR analysis was used, and the results are shown in Figure 3 and Figure 4. In Figure 3a, the main peaks of pure PVA were observed at 3270.68 cm^−1^, 2906.20 cm^−1^, 1417.42 cm^−1^, 1324.86 cm^−1^, 1085.73 cm^−1^, 916.02 cm^−1^, and 835.03 cm^−1^. The largest peak at 3270.68 cm^−1^ is linked to the stretching of O–H from the intramolecular and intermolecular hydrogen bonds [45]. The peak at 2906.20 cm^−1^ is related to the stretching of C-H bond vibrations [46,47]. Another peak at 1417.42 cm^−1^ is linked to CH_2_ bending [36], and the peak at 1324.86 cm^−1^ is linked to C-H deformation [45]. Other peaks at 1085.73 cm^−1^, 916.02 cm^−1^, and 835.03 cm^−1^ were related to C=O stretching, CH_2_ rocking, and C–C stretching, respectively [48]. The spectrum of PVA/S scaffolds had similar FT-IR spectra to the 13% PVA scaffolds (Figure 3). On the other hand, some shifts were observed due to the interactions of the elements after the addition of starch to the pure PVA scaffold. The spectrum of 13% PVA scaffolds shifted from 2906.2 cm^−1^ to 2908.13 cm^−1^, 1417.42 cm^−1^ to 1415.49 cm^−1^, and from 835.03 cm^−1^ to 836.95 cm^−1^. Additionally, a wide peak in the PPE: PSE was observed at 3270.68 cm^−1^ due to stretching vibration of N=H and O=H groups, and the peak at 1637.27 cm^−1^ can be linked to the presence of phenolic components in the extracts and bond stretching in the C=H, C=O, C=C-C=O, C=C, and C-C groups [49,50]. In Figure 4, the bands of the scaffolds, including PPE: PSE extracts, have been shifted from 3266.82 cm^−1^ to 3270.68 cm^−1^ when compared to the 13% PVA/2% S scaffold, and the homogeneity and chemical characteristics of all blended scaffolds were investigated using the shift changes in the bands of the FT-IR spectrum.

The thermal characteristics of the scaffolds were investigated using DSC analysis, and the results are shown in Figure 5 and Figure 6. The glass transition temperature of 13% PVA was observed at around 90 °C, and the melting temperature point was detected at approximately 230 °C [51]. A very slight difference was observed with the addition of starch to the pure PVA scaffold. In addition, the presence of T_g_ point indicates the blend’s homogeneity, and in this study, T_g_ point was observed only in 13% PVA, 13% PVA/1% S, 13% PVA/2% S, and 13% PVA/3% S scaffolds, as shown in Figure 5, which proves that PVA and starch in the scaffold were homogenously blended and printed. Additionally, the T_m_ point was observed in 13% PVA and 13% PVA-based scaffolds. The T_m_ point helps to investigate the crystallization of PVA polymers, and the T_m_ point has been detected in each scaffold due to the high crystallinity of the PVA polymer in each scaffold [52]. DSC curves of the scaffolds that include the PPE: PSE extract shifted and improved the T_m_ point of the 13% PVA compared to the 13% PVA/2% S scaffold (Figure 6). In addition, in Figure 6, it can be seen that there was no significant change in the T_g_ and T_m_ values after the addition of starch or PPE: PSE extract. Still, thermal characteristics of the pure PVA scaffold were improved with the addition of starch and PPE: PSE extract.

The HPLC-DAD methodology was used to determine the qualitative and quantitative phenolic components of pomegranate seed and peel extracts (Figure 7a,b). Linear regression analysis was performed to establish the relationship between peak area and seed and peel extract concentration. In this study, two phenolic compounds were determined in extracts of the plant and using the LOD and LOQ values of the method. Their quantitative amounts were determined and are given in Table 2. Fumaric and quinic acids in pomegranate juice extract were analyzed as 17.56 μg analyte/mg extract and 18.79 μg analyte/mg extract, respectively. The fumaric and quinic acids in pomegranate peel extract were quantitatively analyzed at concentrations of 26.95 μg analyte/mg extract and 33.79 μg analyte/mg extract, respectively. These findings reveal that pomegranate peel extract contains quantities of higher fumaric and quinic acids than pomegranate juice. In addition, the peaks can be identified, according to the literature, according to retention time. The retention time of PSE was around 12, which might be due to the kaempferol component, one of the components of pomegranate flavonoids [53].

The mechanical properties of the scaffolds were tested using the uniaxial tensile testing method, and the results are demonstrated in Table 3. The tensile strength of the 13% PVA/2% S scaffold showed the highest tensile strength among all scaffolds, and with the addition of starch to 13% PVA, the tensile strength of the scaffolds increased from 10.5 ± 2.3 MPa to 23.9 ± 4.1 MPa. For that reason, 13% PVA/2% S scaffolds were chosen to be applied as a control in each characterisation test, especially to observe the starch effect on human gingival fibroblast cells. However, the tensile strength of the scaffolds did not increase gradually as the starch amount in the scaffold increased. Still, when the starch concentration was highest, the tensile strength was higher than pure PVA, and the scaffold was fabricated at the lowest starch concentration. Although no significant difference was observed, the tensile strengths of the 13% PVA/1% S and 13% PVA/3% S scaffolds were measured at 15.1 ± 3.4 and 19.4 ± 2.8 MPa, respectively.Additionally, the PPE: PSEaddition also affected the tensile strength of the scaffolds. The tensile strength of 5% PE scaffolds showedthe highest tensile strength compared to other scaffolds, only including PPE: PSE extract. The highest tensile strength value of the 5% PE scaffold was obtained at 15.8 ± 1.3. When the PPE: PSE concentration was highest, the lowest tensile strength was measured at 8.7 ± 2.5. Therefore, it demonstrates that duee to the liquid form of the PPE: PSE, the mechanical properties of the scaffolds directly were affected and decreased compared to 13% PVA/2% S scaffolds except for the 5% PE scaffold.

The release of PPE: PSE from each scaffold was investigated in PBS (pH:7.4), and the standard calibration was obtained from the PPE: PSE stock in the concentrations of 0.1, 0.2, 0.4, 0.8, and 1 μg/mL, as shown in Figure 8a. The absorbance value of PPE: PSE was detected at 236 nm. The absorbance graph is given in Figure 8b. Figure 8c represents the cumulative release of PPE: PSE from the scaffolds with different amounts of PPE: PSE. According to the results, the highest release of PPE: PSE occurred in the 13% PVA/2% S/11% PE scaffold for 15 min and 7 days. According to the graph in Figure 8c, the 13% PVA/2% S/11% PE scaffold released 50% of the PPE: PSE from itself in the first 15 min. The release of PPe: PSE in the first 15 min might be caused by the presence of PPE: PSE on the surface of the scaffolds. After the first 15 min, a controlled release of PPE: PSE occurred slowly, and the release of PPE: PSE from all scaffolds reached approximately 100% within 7 days. The 13% PVA/2% S/3% PEscaffold had a 50% cumulative release in 5 h. The 13% PVA/2% S/5% PE scaffold released 50% of its PPE: PSE in 3 h. The 13% PVA/2% S/7% PE and 13% PVA/2% S/9% PE scaffolds released 50% of their PPE: PSE within 24 h. According to the study performed by Hussein et al., approximately 80% of the pomegranate extract was completely released from the scaffolds in 160 h [54]. In our study, similar results were obtained, which showed that a 100% release of PPE: PSE was detected after 168 h. In another study carried out by Sadek et al., a 50% release of pomegranate extract was observed within 72 h [55]. These results reported that the release kinetics of the pomegranate extract in our study mainly showed similar behaviour to that reported in the literature. Since the dental membrane should protect and cover the injured area, it should also prevent periodontal pathogen growth in order to decrease complex inflammation with the release of biomolecules within the dental membrane. As a result, the drug release kinetics of this study demonstrated that the fabricated 3D-printed scaffold allowed for the release of antimicrobial pomegranate seed and peel extracts at various time intervals. However, the duration of the PPE:PSE release in each scaffold needs to be improved and should remain in the injured area for a longer period of time.

In the MTT assay, two control groups and five sample groups were tested to observe starch and PPE: PSE effect on human gingival fibroblast cells. However, the calculations were analyzed according to 13%PVA/2% S control group results due to them having the same starch amount in the scaffolds, including PPE: PSE. Therefore, the effects of the starch and PPE: PSE were analyzed, and the results can be seen in Figure 9a. The proliferation of the cells on C1 (13% PVA) and S3 (13% PVA/2% S/7% PE) scaffolds showed decreased behaviour for the first 3 days. It was determined that the addition of starch positively affected the growth of the cells, and a higher proliferation of cells was obtained compared to the 13% PVA scaffold. This result was observed due to the presence of starch in the scaffold, which increased the biocompatibility of the scaffolds. However, a high PPE: PSE content did not show high biocompatibility compared to the control group. Although the PPE: PSE extract did not increase the biocompatibility of the scaffolds, the cell proliferation on the scaffolds containing PPE: PSE was still higher. It was observed that when the amount of PPE: PSE increased, it had a negative impact on HGF cells. Furthermore, for all treatment groups, on the seventh day, the cellular proliferation decreased due to these groups reaching the confluency, and only a small increase in cell proliferation was observed in the sample of S1 (13% PVA/2% S/3% PE). As a result, it can be said that the cell proliferation on treated scaffolds resulted in high compatibility.

There were two control groups and five sample groups in the cell viability test, just as in the cell culture test, and the results are given in Figure 9b. The cell viability in the C2 group (13% PVA/2% S) was higher than in the 13% PVA scaffold because of the starch’s biocompatibility effect on the cells. The addition of starch resulted in a more compatible environment for cells to grow on the scaffolds. However, the viability of the cells on 13% PVA/2% S scaffolds did not decrease or increase, and the viability of the cells stayed the same as on the third and seventh days. Moreover, the highest cell viability occurred on the 13% PVA/2% S/3% PE scaffolds. The cell viability on 13% PVA/2% S/7% PE scaffolds resulted in higher viability than the control group and other scaffolds for the first day, but cell viability on each scaffold on the third day decreased, except for the 13% PVA/2% S/9% PE scaffolds. There was no significant decrease in cell viability on the third day compared to the first and seventh days. Furthermore, the cell viability decreases on the third day but increases on the seventh day in contrast to proliferation results due to the exponential cell growth. Nevertheless, the cell viability of each scaffold (except 13% PVA on the third day and 13% PVA/2% S/9% PE on the seventh day) resulted in a higher than 75% proliferation, which is considered to be non-toxic according to GB/T 16886.5–2003 (ISO 10993–5: 1999) [56]. Therefore, all scaffolds can be considered to be biocompatible with non-toxic contents.

Fluorescence imaging was performed to confirm the MTT assay results and investigate the cellular distribution on each scaffold after 1 day and 3 days of incubation. In Figure 10, the fluorescence images are given. In each image, the cells were located on the scaffolds and the cellular morphologies were apparent on the scaffolds. As expected, the number of cells found on the scaffolds increased with time due to the biocompatibility of the scaffold and the provision of a suitable environment for HGF cells. It was seen that there was no significant difference in the shape and amount of HGF cells located on the 13% PVA and 13% PVA/2% S scaffolds. In addition, the HGF cells located on each scaffold increased on the third day of treatment compared to the first day of incubation, and the HGF cells on all scaffolds showed a typical and similar shape. The amount of located cells on the 13% PVA/2% S/3% PE scaffold had the highest cell viability, and the 13% PVA/2% S/11% PE scaffold also showed higher cell viability than other scaffolds. The number of cells on the 13% PVA/2% S/11% PE scaffold was nearly the same as the number of cells on the 13% PVA/2% S/3% PE scaffold. Moreover, PI staining showed that cellular apoptosis was not observed for each scaffold. However, cells only on the 13% PVA/2% S/7% PE and 13% PVA/2% S/9% PE scaffolds had lower cellular biocompatibility than the control and other scaffolds due to the different cellular morphologies. Since polymeric materials can be used in dentistry, based on the MTT and fluorescence imaging results in this study, it is possible to suggest that a combination of PVA, starch, and PPE: PSE for the fabrication of the dental membrane can be highly preferable for better cell adhesion, differentiation, migration, and attachment due to their high biocompatibility [57]. Additionally, 3D-printed scaffolds were easily able to mimic the extracellular matrix to maintain higher cell viability, and even the structure of the dental membrane can be easily modified according to the injured area’s shape [58].

The scaffolds were tested with three different microorganisms, *S. aureus* (ATCC^®^ 6538™), *C. albicans* (ATCC^®^ 5314™), and *E. faecalis* (ATCC^®^ 29212™), which were chosen due to their presence in the specific location under the dental gum [59,60,61]. As shown in Figure 11 and Table 4, an antimicrobial effect was observed on *S. aureus* and *E. faecalis*. Each scaffold showed a different inhibition zone diameter according to its contents. With the addition of starch into the 13% PVA, the antimicrobial zone on *S. aureus* decreased to 7 mm compared to the 13% PVA scaffold (9 mm). Even though there was no significant difference in the inhibition zone diameter between the 13% PVA scaffold and the 13% PVA/2% S scaffold, the antimicrobial effect increased for *E. faecalis* when starch was added to the scaffold. The 13% PVA/2% S/3% PE and 13% PVA/2% S/7% PE scaffolds had noticeable antimicrobial activity on *S. aureus* and *E. faecalis.*

On the other hand, 13% PVA/2% S/9% PE and 13% PVA/2% S/11% PE scaffolds did not show increased antimicrobial activity compared to the 13% PVA/2% S/3% PE, 13% PVA/2% S/5% PE and 13% PVA/2% S/7% PE scaffolds. Moreover, there was no inhibition zone around the 13% PVA/2% S/9% PE and 13% PVA/2% S/11% PE scaffolds against the *E. faecalis*. According to the results, it was noticed that pomegranate seed and peel extracts have a noticeable antimicrobial activity on both *S. aureus* and *E. faecalis*. It can be concluded that PPE: PSE has a significant antimicrobial effect. However, further research needs be conducted on a large quantity of infections to determine how exactly PPE: PSE impacts species.

## 4. Conclusions

The aim of this research was to investigate the antimicrobial effect of pomegranate seed and peel extracts for dental membrane applications. In this study, an effective dental membrane scaffold was fabricated using 3D printing technology and using the combination of different types of materials, namely PVA, starch, and pomegranate peel and seed extracts. The 3D-printed dental membrane scaffold showed encouraging properties with an appropriate design and antimicrobial effect due to addition of the pomegranate peel and seed extracts. The pore sizes of the dental membrane scaffold varied between 155.86 μm and 280.96 μm without any plugging problems. The pore size used for our study was chosen to be similar to the pore sizes of commercial dental membrane scaffolds. The chosen pore size is highly preferable for mimicking the extracellular matrix for cells. With the addition of the starch and PPE: PSE, the chemical, mechanical, thermal, and physical properties of the scaffolds were improved. The addition of starch and PPE: PSE also increased the mechanical strength and biocompatibility properties of the scaffolds. The highest mechanical strength occurred in the 13% PVA/2% S scaffold. The most favorable and omptimal properties were obtained for the 13% PVA/2% S/3% PE scaffold, especially in terms of antimicrobial activity. The drug release kinetics of pomegranate extract from each scaffold were successfully examined, and the results showed an accomplished cumulative drug release for approximately 7 days, whereby each scaffold might retain their antimicrobial activity for 7 days. However, additional antimicrobial testing should be performed to understand the possibility of the growth and resistance of periodontal pathogens in the affected area. Moreover, the MTT assay and fluorescence imaging results revealed that the 13% PVA/2% S/3% PE scaffold had the highest cell viability and proliferation property compared to other scaffolds, yielding a 106% viability. On the other hand, further experiments, especially clinical experiments, should be performed on our fabricated 3D-printed dental membranes to understand whether each scaffold including PPE: PSE will lead to periodontal recovery and regeneration. Furthermore, scaffolds containing PPE and PSE should be tested on a larger number of periodontal pathogens to determine their antimicrobial effects. However, we believe that the research shows that it has high potential to be used in a clinical setting.

## Figures and Tables

**Figure 1 pharmaceutics-15-00737-f001:**
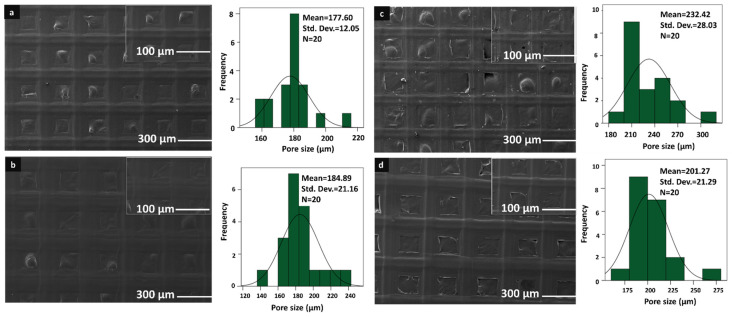
SEM images and histograms of the 13% PVA (**a**), 13% PVA/1% S (**b**), 13% PVA/2% S (**c**), and 13% PVA/3% S (**d**).

**Figure 2 pharmaceutics-15-00737-f002:**
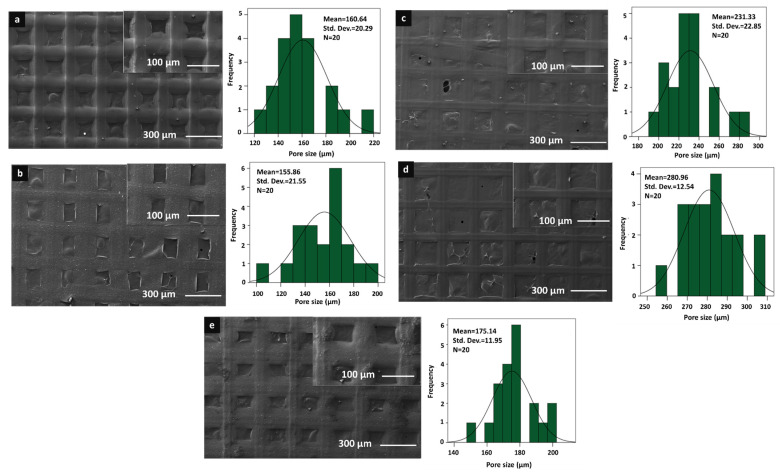
SEM images of the 13% PVA/3% PE (**a**), 13% PVA/5% PE (**b**), 13% PVA/7% PE (**c**), 13% PVA/9% PE (**d**), and 13% PVA/11% PE (**e**) scaffolds.

**Figure 3 pharmaceutics-15-00737-f003:**
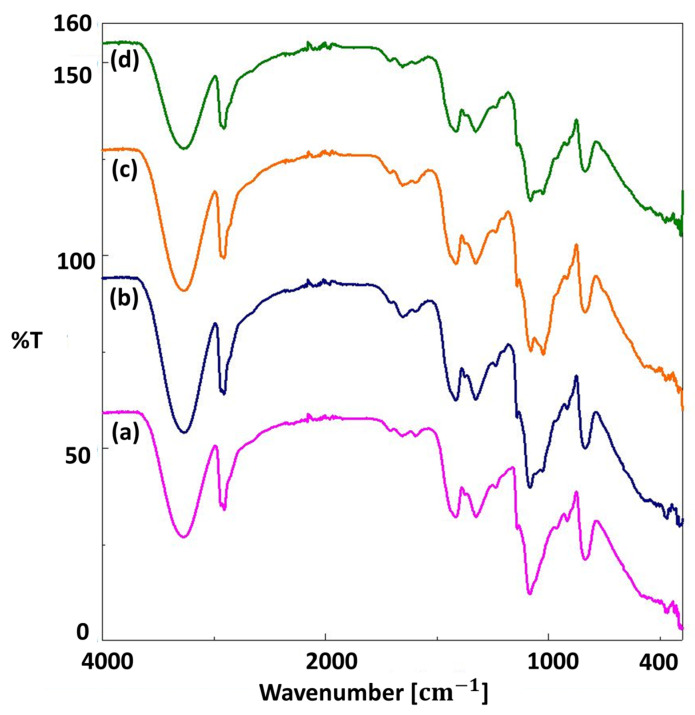
FTIR spectra of the 13% PVA (**a**), 13% PVA/1% S (**b**), 13% PVA/2% S (**c**), and 13% PVA/3% S (**d**).

**Figure 4 pharmaceutics-15-00737-f004:**
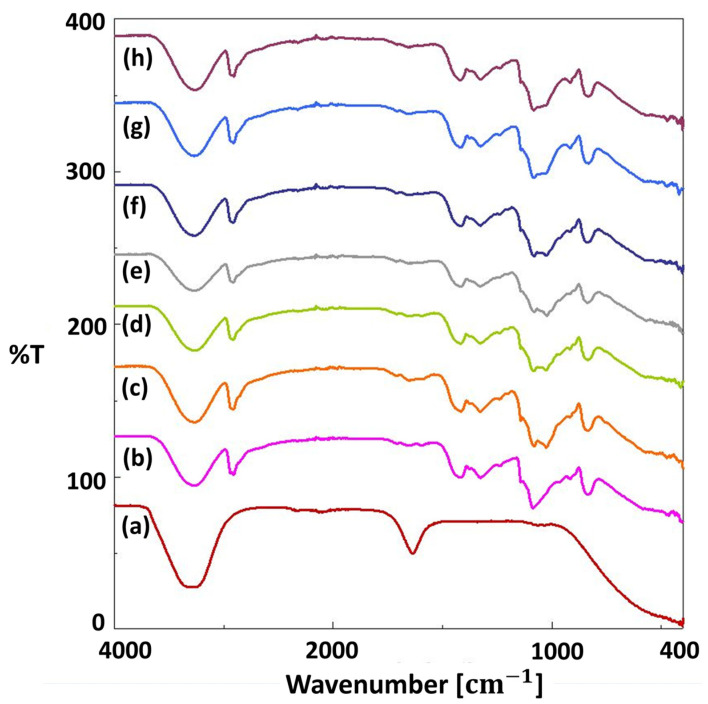
FTIR spectra of the PSE: PPE (**a**), 13% PVA (**b**), 13% PVA/2% S (**c**), and 13% PVA/2% S/3% PE (**d**), 13% PVA/2% S/5% PE (**e**), 13% PVA/2% S/7% PE (**f**), 13% PVA/2% S/9% PE (**g**), and 13% PVA/2% S/11% PE (**h**).

**Figure 5 pharmaceutics-15-00737-f005:**
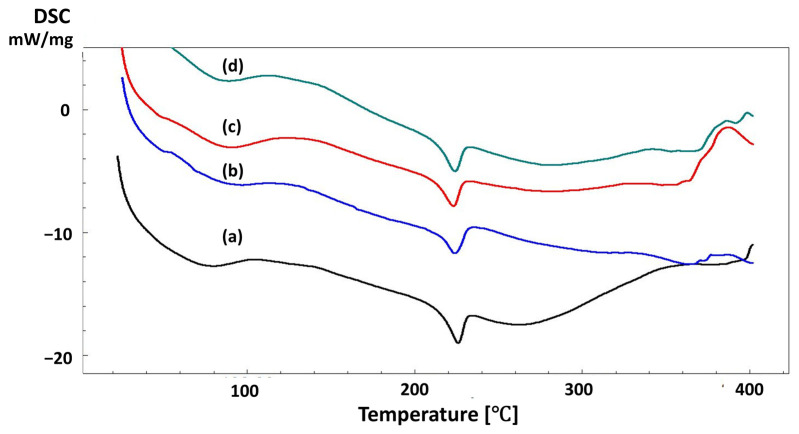
DSC curves of the 13% PVA (**a**), 13% PVA/1% S (**b**), 13% PVA/2% S (**c**), and 13% PVA/3% S (**d**).

**Figure 6 pharmaceutics-15-00737-f006:**
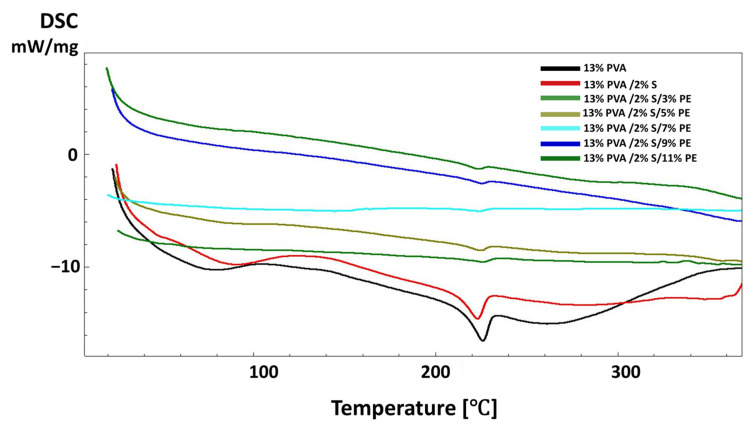
DSC curves of the control and PPE: PSE extracts added scaffolds.

**Figure 7 pharmaceutics-15-00737-f007:**
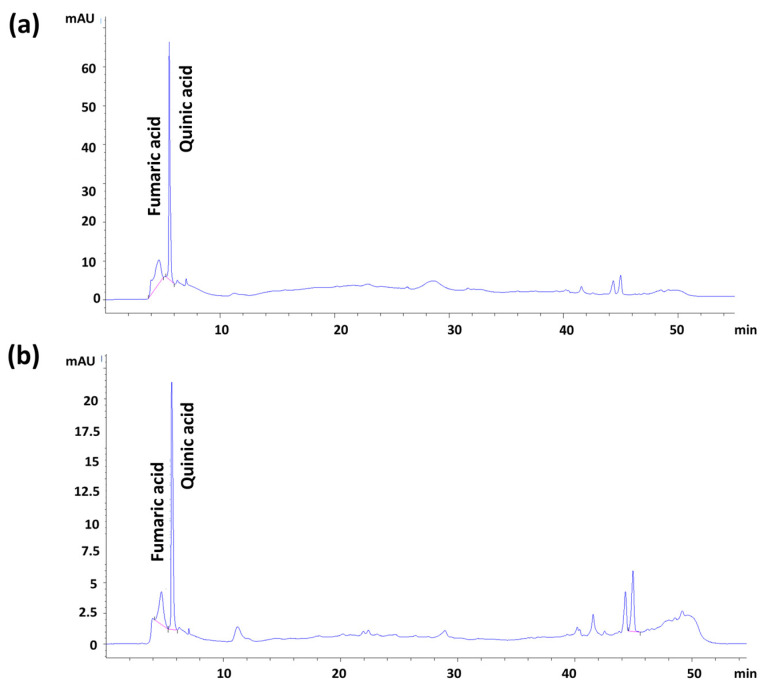
The HPLC-DAD chromatogram of pomegranate peel (**a**) and seed (**b**) extracts.

**Figure 8 pharmaceutics-15-00737-f008:**
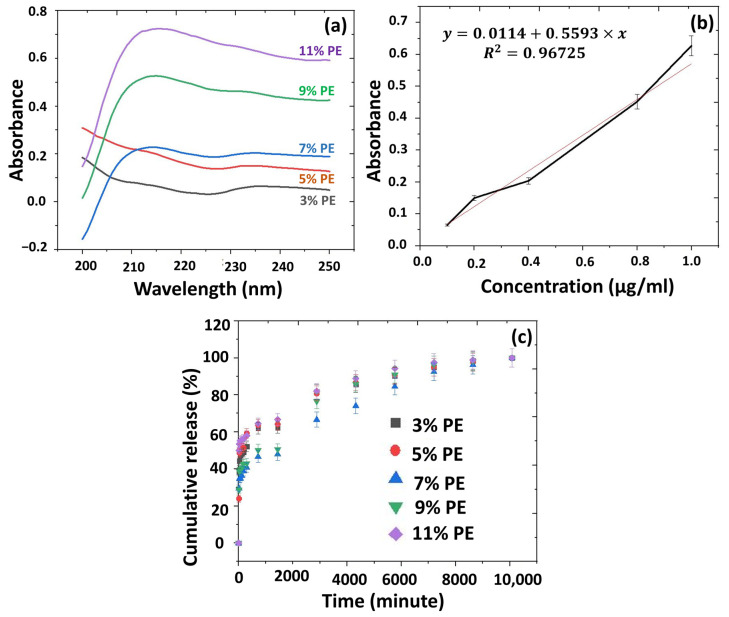
The calibration curve (**a**) and absorbance (**b**) graphs of the PPE: PSE; the cumulative release graph of the extracts from the scaffolds (**c**).

**Figure 9 pharmaceutics-15-00737-f009:**
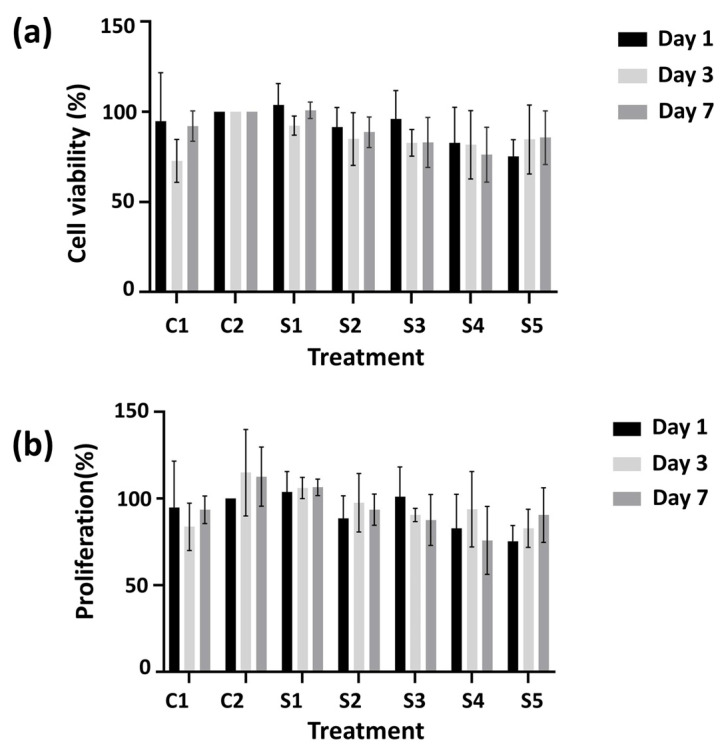
The proliferation (**a**) and viability (**b**) graphs of the control groups and scaffolds. C1: 13% PVA; C2: 13% PVA/2% S; S1: 13% PVA/2% S/3% PE; S2: 13% PVA/2% S/5% PE; S3: 13% PVA/2% S/7% PE; S4: 13% PVA/2% S/9% PE; and S5: 13% PVA/2% S/11% PE.

**Figure 10 pharmaceutics-15-00737-f010:**
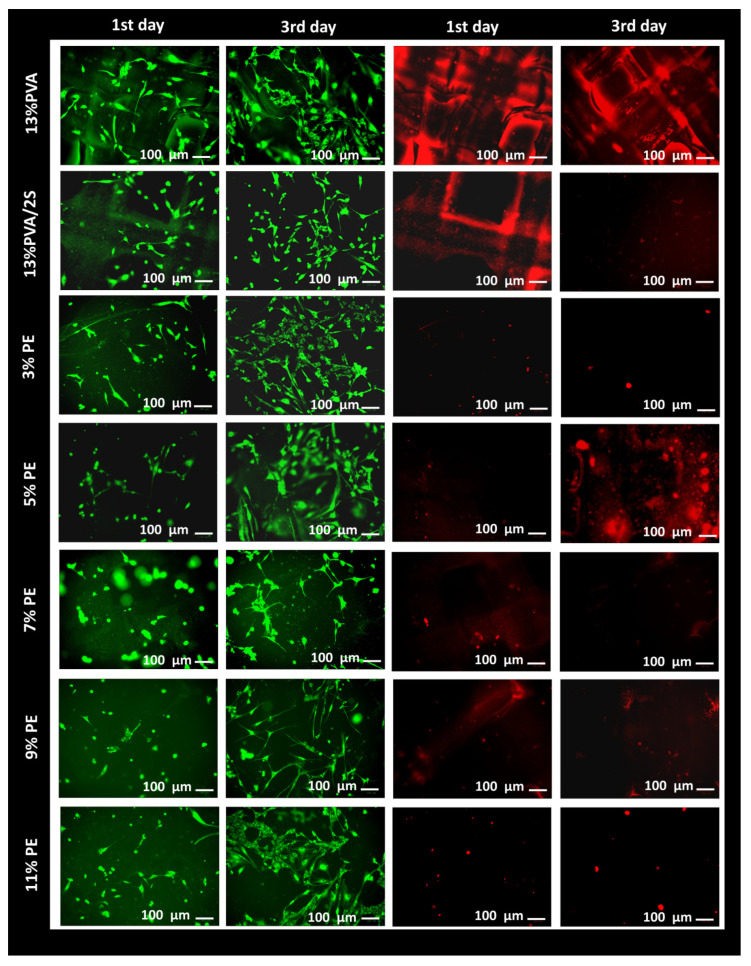
Fluorescence images of the scaffolds after 1 and 3 days of incubation.

**Figure 11 pharmaceutics-15-00737-f011:**
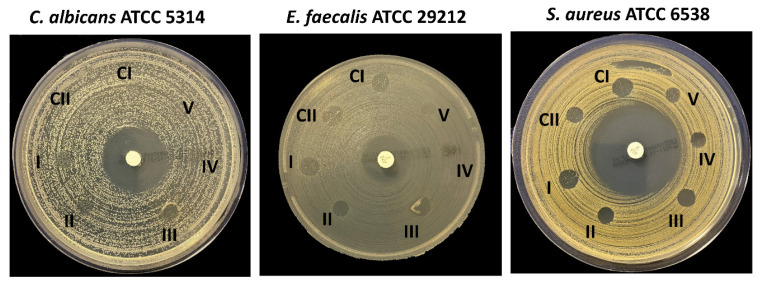
The antimicrobial activity results of the control groups and PPE: PSE scaffolds. CI: Control 1; CII: Control 2; I: 13% PVA/2% S/3% PE; II: 13% PVA/2% S/5% PE; III: 13% PVA/2% S/7% PE; IV: 13% PVA/2% S/9% PE; V: 13% PVA/2% S/11% PE.

**Table 1 pharmaceutics-15-00737-t001:** Physical properties of each solution and pomegranate extracts.

Solutions	Contents	pH Value	Density (g/mL)
13% PVA	13% PVA (*w*/*v*)	6.83	1.038
13% PVA/1% S	13% PVA (*w*/*v*)/1% S (*w*/*v*)	6.22	1.030
13% PVA/2% S	13% PVA (*w*/*v*)/2% S (*w*/*v*)	5.72	1.028
13% PVA/3% S	13% PVA (*w*/*v*)/3% S (*w*/*v*)	5.49	1.021
3% PE	13% PVA (*w*/*v*)/1% S (*w*/*v*)/3% PPE:PSE (*v*/*v*)	5.34	1.015
5 % PE	13% PVA (*w*/*v*)/1% S (*w*/*v*)/5% PPE:PSE (*v*/*v*)	5.21	1.028
7% PE	13% PVA (*w*/*v*)/1% S (*w*/*v*)/7% PPE:PSE (*v*/*v*)	4.98	1.025
9% PE	13% PVA (*w*/*v*)/1% S (*w*/*v*)/9% PPE:PSE (*v*/*v*)	4.84	1.021
11% PE	13% PVA (*w*/*v*)/1% S (*w*/*v*)/11% PPE:PSE (*v*/*v*)	4.77	1.011
PPE:PSE	Pomegranate peel extract:Pomegranate seed extract (1:1)	3.53	1.019
PSE	Pomegranate seed extract (PSE)	3.48	1.024
PPE	Pomegranate peel extract (PPE)	3.63	1.001

**Table 2 pharmaceutics-15-00737-t002:** Quantitative determination of two phenolic compounds using HPLC-DAD.

Compounds	Regression Equation	R^2^	Linear Range (μg/mL)	LOD (μg/mL)	LOQ (μg/mL)
Fumaric acid	y = 27.52x − 43.96	0.9821	20–100	4.25	14.60
Quinic acid	y = 28.79x + 98.34	0.9762	20–100	7.15	10.36

**Table 3 pharmaceutics-15-00737-t003:** The tensile strength and strain values of the 3D-printed scaffolds.

Sample Groups	Tensile Strength (MPa)	Strain at Break (%)
13% PVA	10.5 ± 2.3	15.2 ± 11.9
13% PVA/1% S	15.1 ± 3.4	6.5 ± 4.1
13% PVA/2% S	23.9 ± 4.1	8.4 ± 5.9
13% PVA/3% S	19.4 ± 2.8	4.5 ± 0.1
13% PVA/2% S/3% PE	10.7 ± 2.8	5.5 ± 0.3
13% PVA/2% S/5% PE	15.8 ± 1.3	5.8 ± 0.2
13% PVA/2% S/7% PE	13.6 ± 0.8	7.8 ± 0.01
13% PVA/2% S/9% PE	12.7 ± 3.7	1.8 ± 3.3
13% PVA/2% S/11% PE	8.7 ± 2.5	6.6 ± 2.5

**Table 4 pharmaceutics-15-00737-t004:** The inhibition zone diameters of the control groups and scaffolds.

Group Number	Scaffolds	*S. aureus* (mm)	*E. faecalis* (mm)	*C. albicans* (mm)
CI	13% PVA	9	7	0
CII	13% PVA/2% S	7	8	0
I	13% PVA/2% S/3% PE	8	7	0
II	13% PVA/2% S/5% PE	8	5	0
III	13% PVA/2% S/7% PE	8	7	0
IV	13% PVA/2% S/9% PE	6	0	0
V	13% PVA/2% S/11% PE	6	0	0
	AMP-Ampicillin (2 µg)	39	20	-
	Chlorhexidine gluconate (20%)	-	-	27

## Data Availability

Data will be available on the request.

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
