# Peer review of "A Novel Approach for the Fabrication of 3D-Printed Dental Membrane Scaffolds including Antimicrobial Pomegranate Extract"

_pharmaceutics, 2023, doi:10.3390/pharmaceutics15030737_

Round 1

Reviewer 1 Report

The authors construct a dental membrane and show high antimicrobial effects on scaffolds including pomegranate extracts mixture. It is a useful and interesting study. However, I believe the article would be better if the author can explain or improve the following puzzles.

  1. The purpose of the article is to cover the damaged area and aid in the healing process. But there are relatively few experimental designs in the paper to validate "aiding the healing process.
  2. In the bar graphs of SEM representation results of scaffolds in Figures 1 and 2, no statistical analysis was performed, and it is unknown whether it was due to insufficient samples?
  3. The evaluation of the mechanical properties of scaffolds is too simple because the application of scaffolds in the oral cavity is complicated, and the compressive and bending resistance of scaffolds are equally important. Therefore, it is suggested to add compression test and other test results.
  4. The bar chart in Figure 9 has no error bars, and the experimental results were not analysed statistically.
  5. There is only one antibacterial ring test in the antibacterial test, which is too simple and does not adequately reflect the antibacterial properties. It is suggested to increase the antibacterial performance evaluation experiment.
  6. The species of oral bacteria are unique, and it is recommended to select specific oral bacteria to evaluate their antibacterial properties.
  7. The ability of scaffolds to allow cell attachment and differentiation is an important property, and it is recommended to increase the evaluation of this property.

Author Response

Thank you for your kind reviews. We have made changes in the manuscript according to your suggestions. 

Reviewer 2 Report

1. The first table in section 3 has no table title and description.

2.  Also, the same table has missing units for the density.

3. The text in figures 1 and 2 are too small. Suggest enlarging the font size.

4. Can the authors explain in the manuscript why 13% PVA was used? were there any optimization tests being conducted?

5. Missing scale bar in figure 10.

6. figure 8 has poor resolution, suggest replacing with a high resolution image.

7. Since this manuscript is about 3d printing for biomedical application, suggest citing and discussing relevant works as suggested below.

a. Soetedjo, A. A. P., Lee, J. M., Lau, H. H., Goh, G. L., An, J., Koh, Y., ... & Teo, A. K. K. (2021). Tissue engineering and 3D printing of bioartificial pancreas for regenerative medicine in diabetes. Trends in Endocrinology & Metabolism32(8), 609-622.

b. Pillai, S., Upadhyay, A., Khayambashi, P., Farooq, I., Sabri, H., Tarar, M., ... & Tran, S. D. (2021). Dental 3D-printing: transferring art from the laboratories to the clinics. Polymers13(1), 157.

c. Dawood, A., Marti, B. M., Sauret-Jackson, V., & Darwood, A. (2015). 3D printing in dentistry. British dental journal219(11), 521-529.

Author Response

(The authors gave the same response as above.)

Reviewer 3 Report

The topic of the present study might be very interesting and clinically relevant.

However, the terminology is incorrect throughout the manuscript (e.g., "skin" intra-orally?). There is no clear identification of the issue and the current gap in the available evidence. The background is not defined. No Clinical applications (also speculations) have been discussed. The discussion section is missing. The limitations and strengths of the study are not detailed. 

I would suggest discussing background with dentists and reorganizing the manuscript.

Some suggestions and comments are highlighted in the file attached.

Author Response

(The authors gave the same response as above.)

Round 2

Reviewer 1 Report

Accept in present form

Reviewer 3 Report

The manuscript has been improved, although I still find it a bit confusing in the structure. The statement in the Introduction section should be corrected "One of the 45 most common problems in dental health care is chronic inflammatory disorders. [1, 2,3]. This inflammatory disorder is also called periodontitis". Please, see also Computed tomography evaluation of jaw atrophies before and after surgical bone augmentation eid=2-s2.0-85081018833.

Author Response

Thank you for your valuable comments and thoughts. The structure of the sentence in the introduction part was corrected and changed where it was relevant according to the reviewer's request.